# Hepatic MicroRNA Expression by PGC-1α and PGC-1β in the Mouse

**DOI:** 10.3390/ijms20225735

**Published:** 2019-11-15

**Authors:** Elena Piccinin, Maria Arconzo, Giusi Graziano, Michele Vacca, Claudia Peres, Elena Bellafante, Gaetano Villani, Antonio Moschetta

**Affiliations:** 1Department of Interdisciplinary Medicine, University of Bari “Aldo Moro”, Piazza Giulio Cesare 11, 70124 Bari, Italy or; 2INBB, National Institute for Biostuctures and Biosystems, 00136 Rome, Italy; arconzomaria92@gmail.com (M.A.); giusi.graziano78@gmail.com (G.G.); claudiaperes18@gmail.com (C.P.); 3Metabolic Research Laboratories, Wellcome Trust-MRC Institute of Metabolic Science, Box 289, Addenbrooke’s Hospital, Cambridge CB2 0QQ, UK; mv400@medschl.cam.ac.uk; 4Fondazione Mario Negri Sud, Santa Maria Imbaro, 66030 Chieti, Italy; elenabellafante@gmail.com; 5Department of Basic Medical Sciences, Neurosciences and Sense Organs, “Aldo Moro” University of Bari, 70124 Bari, Italy; gaetano.villani@uniba.it

**Keywords:** liver metabolism, liver diseases, coactivators, PGC-1, microRNA

## Abstract

The fine-tuning of liver metabolism is essential to maintain the whole-body homeostasis and to prevent the onset of diseases. The peroxisome proliferator-activated receptor-γ coactivators (PGC-1s) are transcriptional key players of liver metabolism, able to regulate mitochondrial function, gluconeogenesis and lipid metabolism. Their activity is accurately modulated by post-translational modifications. Here, we showed that specific PGC-1s expression can lead to the upregulation of different microRNAs widely implicated in liver physiology and diseases development and progression, thus offering a new layer of complexity in the control of hepatic metabolism.

## 1. Introduction

Variations of environmental conditions require a constant but flexible adaptation of an organism. In particular, the rapid changes occurring between fed and fasted state are fulfilled by enzymes controlling catabolic pathways, while the fast-fed transition is characterized by modulation of anabolic programs. An efficient fed-fast cycle is of fundamental importance, since failure to induce or inhibit the expression of metabolic genes has been associated with the development of diseases [1,2]. In the cell, this is primarily achieved through the activation of a transcriptional machinery that leads to the expression of genes with different metabolic functions.

The members of the peroxisome proliferator-activated receptor (PPAR)-γ coactivators (PGC-1s) family are key players in the regulation of energy metabolism. Usually upregulated in conditions of increased energy demands, these coactivators modulate the expression of several genes involved in mitochondrial biogenesis and function, as well as in other catabolic and anabolic pathways [3]. The liver is probably the organ that best illustrates the different contributions of the members of the PGC-1 family to guarantee whole-body homeostasis. Even though many studies have been carried out considering their specific implication on liver metabolism, several controversies about the specific functions of different PGC-1 family members still remain.

Another level of regulation in the fed-fast transition is represented by microRNA (miRNA), small noncoding RNA molecules which control gene expression post-transcriptionally, via complementary base pairing with mRNA. These microRNAs are not only able to orchestrate system-level control over liver physiology and whole-body energetics, but can also contribute to the pathogenesis of many liver diseases [4,5]. Intriguingly, miRNAs can modulate the activity of PGC-1s, thus offering a new level of complexity in the regulation of metabolic homeostasis [4]. Conversely, to our knowledge, little is known about the regulation of miRNAs expression by these coactivators.

The present review discusses some of the new findings regarding the control of PGC-1s expression and activity as well as the ability of PGC-1s to induce microRNA variously involved in liver physiology and pathophysiology.

## 2. Results and Discussion

### 2.1. The Proliferator-Activated Receptor (PPAR)-γ Coactivator (PGC-1) Family of Transcriptional Coactivators

The PGC-1 family comprises powerful regulators of cellular programs, activated in response to a plethora of conditions, such as fasting, physical exercise and thermogenesis that collectively require increased energy demand [3]. PGC-1α is the most well-studied member of this coactivator’s family, firstly identified as transcriptional coregulator of PPAR-γ in brown adipose tissue [6]. Then, PGC-1β and PRC (PGC-1 related coactivator) were also recognised, thanks to the extensive sequence homology and analogous biological functions [7,8]. However, whereas PRC is ubiquitously expressed in the whole body, PGC-1α and PGC-1β share a similar expression pattern. Indeed, both PGC-1α and PGC-1β are thoroughly induced in tissues with high oxidative metabolism, like the heart, brown adipose tissue and type I muscle fibres [9,10,11]. Still, PRC function has not been well explored, by contrast with the other two members of the family. In particular, PRC is a growth-regulated cofactor, with the characteristic of an immediate early gene, i.e., gene expressed early in the cell growth program as a primary response to multiple stimuli [12]. As expected for a cell growth controller, silencing of PRC results in a decreased cell proliferation mainly attributable to abnormal mitochondria formation [13].

A critical aspect of PGC-1s is their highly versatile capability to interact with different transcription factors, activating distinct biological programs in various tissues. PGC-1α was originally described through its functional interaction with PPARγ in brown adipose tissue (BAT), a mitochondria rich tissue, where it regulates adaptive thermogenesis [6]. Further studies revealed that the PGC-1s carry out several biological responses finalized to equip the cell to cope with the alterations of energy demands, including mitochondrial biogenesis, cellular respiration capacity and substrate uptake and utilization.

The stimulatory effects of PGC-1α on mitochondrial functions are achieved mainly through its coactivation of nuclear respiratory factors 1 and 2 (NRF1 and NRF2, respectively) and the estrogen-related receptor α (ERRα) [10,14]. NRFs regulate the expression of mitochondrial transcription factor A (TFAM), a nuclear-encoded transcription factor essential for replication, maintenance, and transcription of mitochondrial DNA [15]. NRF1 and NRF2 also control the expression of genes coding for mitochondrial components of the oxidative phosphorylation apparatus, such as ATP synthase, cytochrome *c*, and cytochrome *c* oxidase [16,17]. Therefore, by stimulating the concomitant expression of both mitochondrial- and nuclear-encoded proteins, PGC-1α and PGC-1β are able to enhance the aerobic bioenergetic capacity of the cell, by boosting fatty-acid β-oxidation, Krebs cycle as well as oxidative phosphorylation.

Besides their well-known role of stimulating mitochondrial biogenesis, PGC-1s also modulate a variety of other cellular bioenergetic processes, thanks to their interaction and potential binding to multiple transcription factors [18,19]. This versatility of PGC-1s underlines their pleiotropic functions involved in the tissue-specific regulation of several metabolic pathways localized in different cellular compartments.

Although the role of PGC-1α and PGC-1β could sometimes appear to be redundant due to their overlapping functions, in the liver these coactivators have the ability to regulate distinct processes that together contribute to hepatic metabolism homeostasis. Indeed, although both coactivators can increase oxidative metabolism and antioxidant response, PGC-1α plays a major role in catabolic pathways, such as gluconeogenesis and fatty acids β-oxidation, by upregulating hepatocyte nuclear factor 4α (HNF4α), forkhead box protein O1 (FOXO1) and glucocorticoid receptor (GR), while PGC-1β drives new lipid synthesis and very low density lipoprotein (VLDL) trafficking via coactivation of liver X receptor (LXR) and sterol regulatory element-binding protein 1c (SREBP1c) [20,21,22,23,24,25].

Hence, in general conditions of stress, such as food shortage, PGC-1s are upregulated and promote metabolic programmes aimed to fulfil the energy requirements. Moreover, their ability to sustain opposite metabolic programmes is essential to prevent futile substrates cycles. Indeed, in a fed status PGC-1β enhances de novo lipogenesis, while in fasting conditions PGC-1α supports gluconeogenesis. In this way, both coactivators are entailed in a fine regulation of hepatic glucose availability.

In this view, alterations of PGC-1s pathways can lead to the onset of hepatic diseases, such as non-alcoholic fatty liver disease (NAFLD), non alcholic steatohepatitis (NASH) and hepatocellular carcinoma (HCC) [18]. However, despite the large plethora of studies carried out on this topic, some issues regarding the PGC-1s’ contribution to liver metabolism and diseases need still to be addressed.

#### Structure and Regulation

All the members of the PGC-1 family share a similar modular structure, with a strong transcriptional activation domain at the N-terminus. The C-terminal region of PGC-1s harbours a Ser/Arg rich domain and an RNA binding domain that couples pre-mRNA splicing with transcription [26]. PGC-1α and PGC-1β share an additional domain of similarity in the internal region, which roughly spans 200 aminoacids and functions as a repression domain [27].

The PGC-1s interact with several powerful histone acetyl transferase (HAT) complexes at their N-terminal region, including cAMP response element-binding protein (CREB)-binding protein, p300, and steroid receptor coactivator-1 (SRC-1) [27]. These proteins acetylate histones and remodel chromatin structure, thus allowing access of additional factors for transcriptional activation. Other activation complexes dock at the C-terminus domain: the TRAP/DRIP (thyroid receptor-associated protein/vitamin D receptor-interacting protein, also known as the Mediator complex) that facilitates direct interaction with the transcription initiation machinery [28] and the SWI/SNF (switch/sucrose non-fermentable) that acts as a chromatin-remodelling complex through its interaction with BAF60a [29].

Thus, the remarkably powerful coactivation capacity of PGC-1s may be based on their property to act as protein docking platforms for the recruitment and assembly of chromatin remodelling and/or histone-modifying enzymes that facilitate access to DNA for the transcription machinery. Furthermore, the PGC-1α transcriptional activator complex is also able to displace repressor proteins such as histone deacetylase and small heterodimer partner (SHP) on its target promoters, providing an alternative mechanism for an intensified gene transcription [30].

PGC-1α and PGC-1β share similar domains and several clusters of conserved amino acids, including the LXXLL motif which is recognized by nuclear receptors and host cell factor 1 interacting motif [7,31].

The mechanism through which the PGC-1s activate gene expression has to date been poorly understood. Initial studies have identified an extremely powerful autonomous transcriptional activity at the N-terminal region, due to the ability of PGC-1s to dock two additional coactivators with acetyl transferase activity on this domain. Similar observations have been made at the C-terminal region interacting with proteins involved in RNA processing and with the TRAP/DRIP complex involved in transcriptional initiation. However, how spatially and temporally all these complexes are assembled to PGC-1s in order to control gene expression is unknown. The major current hypothesis is that PGC-1s bind to a specific transcription factor at promoter regions, followed by the recruitment of P300 and TRAP/DRIP complexes which open the chromatin through histone acetylation activity, thus allowing the initiation of the transcription through RNApolII. Moreover, the presence of several proteins involved in RNA elongation and processing in the PGC-1s supercomplex suggests that it might move along the elongating RNA and take part in the mRNA maturation. To terminate gene expression, GCN5 (general control of amino acid synthesis), an acetyl transferase, acetylates PGC-1s at several lysine residues, causing a re-localization of PGC-1s from the promoter region to subnuclear foci where its transcriptional activity is inhibited [32,33]. By contrast, the enzyme SIRT1 (Sirtuin 1) activates PGC-1s by deacetylating lysine residues, thus leading to expression of PGC-1s target genes [34]. Intriguingly, both GNC5 and Sirt1 act as energy sensors of the cell. When energy levels are low, AMPK (AMP-activated protein kinase), by increasing NAD^+^ levels, promotes the deacetylase activity of SIRT1, which results in PGC-1s activation and increased mitochondrial biogenesis and function. Contrarily, in caloric excess conditions, GCN5 inhibits PGC-1s by acetylation, thus lowering new mitochondria formation [35]. Thereby, via reversible acetylation/deacetylation of PGC-1s, Sirt1 and GNC5 give a tightly but flexible regulation of the mitochondrial synthesis in order to fulfil the cellular energy demand.

Another level of complexity is introduced by additional PGC-1s post-translational modifications, for instance phosphorylation and methylation as well as by the interaction with other proteins, such as corepressors which alter PGC-1s stability and activity. PGC-1α could be phosphorylated directly by three different kinases. p38 mitogen-activated protein (MAP) Kinase phosphorylates PGC-1α in its repression domain, allowing a more stable and active protein [36,37]. AMPK could also phosphorylate PGC-1α, leading to an increase of its target gene expression activity [38]. Notably, this phosphorylation is required for the subsequent SIRT1-mediated deacetylation, thus suggesting a convergent biological effect directed at increasing the specificity of PGC-1α activation [39].

By contrast with the aforementioned kinases, insulin-dependent activation of AKT (Protein Kinase B) results in a more unstable PGC-1α protein, with a lower transcriptional activity [40]. Once activated, AKT can exert both a direct and an indirect action on PGC-1α, the latter by phosphorylating and stabilizing the Clk2 protein kinase. The Clk2-dependent phosphorylation of PGC-1α blunts its coactivator activity leading to decreased glucose production due to the repression of gluconeogenesis program [41].

Additionally, GSK3β (glycogen synthase kinase 3β) can phosphorylate PGC-1α in order to promote its intranuclear proteasomal degradation during transient conditions of oxidative stress. However, a chronic stress signal overcomes the GSK3β-mediated degradation, thus resulting in the increased PGC-1α expression [42]. Furthermore, PGC-1α is a target of PRMT1 (protein arginine N-methyltransferase 1). PRMT1 methylates PGC-1α on three arginine residues in the C-terminus, hence promoting its activation [43].

By contrast with PGC-1α, PGC-1β post-translational modifications have been poorly studied. It has been observed that SYVN1 (Synoviolin), an E3 ubiquitin ligase, ubiquitinates PGC-1β and, consistently, SYVN1 deficient mice displayed decreased body weight and upregulation of PGC-1β and of its target genes regulating mitochondrial biogenesis and function [44]. Overall, it is plausible that PGC-1β acts in multiple protein complexes whose composition might depend on the specific target genes as well as on the composition of different metabolic signals.

### 2.2. MicroRNA

MicroRNAs (miRNAs) are small non-coding RNAs, widely involved in the regulation of biological processes, such as proliferation, development and energy metabolism. Firstly described in 1993, microRNAs are small noncoding RNAs, about 22 nucleotides long that can finely tune gene expression, either reducing transcript levels or interfering with protein translation [45,46,47]. Therefore, the discovery of microRNAs deepens our knowledge on the post-transcriptional modifications of gene expression and on how this process can regulate metabolic homeostasis and diseases development.

Genes that encode miRNAs are firstly transcribed by RNA polymerase II from DNA into an initial hairpin transcript, called pre-miRNA, via either a canonical pathway, in which they are cropped by the Drosha-DGCR8 complex, or via the Mirtrons pathway. Pre-miRNA are then exported into the cytoplasm, where they are processed by Dicer into a mature, single stranded miRNA. Finally, miRNAs are loaded onto the RNA-induced silencing complex (RISC) and guided to their mRNA targets through interactions with members of the Argonaute family. The unique complementarities between miRNA and mRNA could reduce or increase the levels of a gene product [48].

The generation of mature miRNAs can be controlled both at transcriptional and posttranscriptional level, and any alteration occurring in their synthesis leads to abnormal physiological and developmental processes that can eventually contribute to disease onset [49,50,51].

In the liver, the distribution and function of some miRNAs is cell-specific and their dysregulation has been associated with hepatic dysfunctions, to the point that extracellular circulating microRNAs released from injured liver could be used as biomarkers of liver disease [5,51]. Hepatic miRNAs can affect glucose and lipid homeostasis, inflammatory programmes as well as cell proliferation and apoptosis. For instance, miR-122, an abundant liver microRNA which accounts for 70% of liver total miRNA, regulates cholesterol biosynthesis and lipid metabolism [52,53]. Low levels of miR-122 are found in non alcholic steatohepatitis (NASH) and hepatocellular carcinoma (HCC), in concomitance with inflammatory infiltrates producing pro-tumorigenic cytokines [54,55]. Mice with total body miR-122 ablation are viable, but present an altered lipid metabolism that results in steatohepatitis, fibrosis and liver cancer development [56]. On the contrary, miR-122 delivery in MYC proto-oncogene, bHLH transcription factor (MYC)-driven HCC mouse model contributes to tumorigenesis inhibition [55]. Finally, increased levels of circulating miR-122 are observed in association with liver injury, thus sustaining the idea that specific microRNAs can be used as markers of liver damage and inflammation [57].

New evidences indicate that, besides ‘canonical’ role of miRNAs in inhibition of target gene expression, microRNAs could exert more complex and different roles. MiRNA can reduce or increase mRNA expression via mRNA deadenylation, translational repression or activation, or other not yet defined mechanisms [51]. Although more studies are required in order to understand the vast complexity of miRNAs regulation, it is now clear that microRNAs participate to the maintenance of the whole-body metabolic homeostasis, and alteration of their expression could not only directly or indirectly modulate tissue metabolism, but can also favour the onset of diseases. In this view, being aware of the genes and relative pathways regulated by microRNAs, but also of the transcriptional factors or coregulators that modulate miRNAs expression is of fundamental importance.

#### 2.2.1. Regulation of PGC-1s by MicroRNAs

The study of microRNAs offers an additional level of complexity on the modulation of transcriptional coactivators activity. Until now, numerous analyses have revealed that different microRNAs can directly control the expression of PGC-1s, thereby having an effect on whole body metabolic homeostasis.

In an elegant study, Maniyadath et al. showed that hepatic-fed microRNAs exert a network-level control over fasting genes to regulate metabolic and mitochondrial pathways, thus avoiding futile cycles [4]. In particular, let7i, miR-221 and miR-222 are the most abundant microRNAs in re-fed liver that are able to act in a convergent and additive way to control gluconeogenesis, fatty acid oxidation and mitochondrial biogenesis by directly targeting PGC-1α [4]. The loss of these microRNAs resulted in the failure to inhibit the expression of PGC-1α and downstream genes, thus contributing to the onset of metabolic dysfunction and liver disease. The downregulation of PGC-1α by miR-19b/221/222 has also been observed in human atherosclerotic vessels, where the coactivator plays a protective role against vascular complications [58]. Likewise, estrogen deficiency leads to diminished PGC-1α expression that results in the formation of compromised mitochondria, eventually leading to enhanced cardiovascular risk in age-related ventricular concentric remodelling. In particular, estrogen negatively regulates miR-23a which could directly bind PGC-1α, thus turning down its expression [59]. It has been also observed in hepatocellular carcinoma that miR-23a can lessen gluconeogenesis programs via direct binding at 3′UTR of both glucose 6-phosphatase and PGC-1α [60]. Similarly, adenoviruses-mediated overexpression of miR-29a-c in primary hepatocytes and mouse liver downregulates PGC-1α, thus contributing to the attenuation of hepatic glucose production and alleviation of hyperglycemia [61].

Moreover, even miR-696 can regulate PGC-1α expression. This microRNA is downregulated in the gastrocnemius during exercise and in the liver of genetically obese mice (ob/ob). Therefore, by decreasing PGC-1α levels, miR-696 contributes to the regulation of muscle mitochondrial biogenesis, hepatic gluconeogenesis and insulin resistance [62,63].

Finally, different studies indicated that miR-34a can directly target PGC-1α in hepatic and adipose tissue. MiR-34a levels rise in the liver and in the white adipose tissue of high-fat diet (HFD) fed mice. When miR-34a is knocked-out in mice using genetical or lentiviral approach, the PGC-1α expression levels raise in liver and white adipose tissue [64,65,66]. Interestingly, activation of FXR (farnesoid X receptor) by selective ligands decreases the expression of miR-34a, which in turn it is not able to downregulate SIRT1, thus leading to enhanced PGC-1α activity [64]. However, despite the marked elevation of the coactivator, the consequences of this miR-34-KO are different between the two mouse models, probably due to the approaches used to shut-down the microRNA. Indeed, the whole-body ablation of miR-34a leads to an increased body weight due to the production of dysfunctional mitochondria that promote lipid uptake and storage, thus resulting in the accumulation of inflammatory markers [66]. On the contrary, lentiviral-mediated transient downregulation of miR-34a reduces adiposity and enhances mitochondrial copy number as well as oxidative capacity in mice fed with diet-induced obesity [65]. Notably, it has been shown that full body PGC-1α knock out mice are lean, but display defective mitochondrial functions [67].

Fewer studies have been published on the microRNA-mediated regulation of PGC-1β. Oleanolic acid, an active component of the traditional Chinese herb *Olea europaea L*., induces the expression of miR-98-5p which directly targets the 3′UTR of PGC-1β and promotes its mRNA degradation. Mice fed with HFD supplemented with oleanolic acid display ameliorated hyperlipidaemia associated with decreased PGC-1β levels, and treatment with miR-98-5p inhibitors reverses this effect [68]. Similarly, in the goat mammary gland tissue, the overexpression of miR-25 represses PGC-1β expression and, consequently, decreases triacylglycerol and lipid accumulation [69].

Generally, these studies highlight that also microRNA contribute to the well calibrated action of both PGC-1α and PGC-1β on metabolic programmes in order to guarantee the whole-body homeostasis.

#### 2.2.2. Regulation of MicroRNAs by PGC-1s

Hepatic microRNAs are involved not only in the maintenance of normal liver functions, but also in liver diseases and cancer. Despite a large number of studies demonstrating that different microRNAs control PGC-1s expression, to date the researches about the regulation of microRNAs expression by PGC-1s are almost missing.

Notably, two microRNAs are embedded in the first intron of the PGC-1β gene, miR-378 and miR-378*, which originate from a common hairpin RNA precursor [70]. Selective deletion of these miRNAs in mice, leaving the *PGC-1β* gene intact, resulted in protection against diet-induced obesity. Overall, miR-378 and miR-378* counterbalance the metabolic actions of PGC-1β by controlling the oxidative capacity and the fatty acid metabolism [71].

Here, we report the data of a miRNAs microarray analysis on hepatic samples collected from fasted transgenic mice in which the coactivators PGC-1α or PGC-1β have been specifically overexpressed in the liver, namely LivPGC-1α and LivPGC-1β, respectively. The two differential microarray analyses versus wild type mice have been carried out for LivPGC-1α mice (Figure 1) and for LivPGC-1β mice (Figure 2). The most significative miRNAs differences emerging from each analysis have been then validated by real-time quantitative polymerase chain reaction (qPCR) (Figure 3) that allowed the comparison of the microRNA expression levels also between the two coactivators. Notably, the genomic location of most of these miRNAs is inside or closed to genes regulated by PGC-1s or closely associated to PGC-1′s activity (Table 1). Therefore, it is plausible that the expression of these microRNAs represents a continuum of the gene regulation committed by PGC-1s. Intriguingly, for most of these microRNAs few literature data are available. Therefore, it will be interesting to explore if they are implicated in the regulation of hepatic metabolic processes or in liver diseases (Figure 4).

Notably, miR-30c-1, miR-677 and miR-345-5p are significatively increased in LivPGC-1β mice with respect to the other two groups, whereas miR-146b and miR-34a are significatively upregulated in LivPGC-1β compared to wild type and LivPGC-1α respectively. Interestingly, miR-30c-1 and mir-677 are enclosed into two genes whose expression is regulated by PGC-1s, i.e., nuclear transcription factor Y subunit gamma (*Nfyc*) and ATP synthase F1 subunit beta (*Atp5b*), respectively (Table 1). Mir-677 has been found to be downregulated in the liver of female mice infected with *Plasmodium chabaudi malaria* and a concomitant upregulation of genes involved in the inflammatory pathway, as interleukin (IL)-1β, tumor necrosis factor-α (TNF-α) and nuclear factor κB (NF-κB), has been observed [72]. Anyway, if miR-677 can play any role in mitigating the inflammatory response has not yet been established.

Mir-345-5p has been described in several types of cancer: in pancreatic ductal adenocarcinoma and in acute myeloid leukaemia miR-345-5p plays a tumor suppressive role, whereas in prostate cancer it promotes cell proliferation and migration by blocking p21, responsible for the inhibition of cyclin-dependent kinases (CDK) CDK1 and CDK2 [73,74,75]. Moreover, in tissue and cell lines of cholangiocarcinoma, the upregulation of the long non-coding RNA (lncRNA) AS1 sponges miR-345-5p, thus removing the inhibition of collagen type VI alpha 3 (Col6A3), direct target of mir-345-5p, and allowing tumor progression [76]. Contrariwise, the comparison between different datasets of cirrhosis patients revealed that miR-345-5p expression, together with other four microRNAs, is associated with increased levels of genes involved in liver fibrosis [77]. Finally, higher circulating levels of miR-345-5p have been recently observed in children with early stage diabetes mellitus type 1 as compared with age-matched controls [78].

In the liver, miR-146b has been widely correlated with inflammation in both NAFLD and HCC [79]. MiR-146b is a negative regulator of toll-like receptor 4 (TLR4) and tumor necrosis factor receptor-associated factor 6 (TRAF6) and its administration prevents liver injury in acute liver graft injury in rats and ameliorates HFD-induced NAFLD in mice [80,81,82]. Analogously, decreased levels of miR-146b have been detected in serum of NAFLD patients and in a mouse model of NAFLD induced by methionine choline deficient diet [82,83]. Intriguingly, ablation of miR-146 in mice leads to severe liver steatosis and hepatitis by the upregulation of TNF-α and IL-6 [82]. Conversely, elevated hepatic level of miR-146b has been found in rats and mice treated with HFD as well as in obese subjects with NAFLD [84,85,86]. Notably, the upregulation of miR-146b in obese individuals correlates with a reduction of glucose metabolism and fatty acids mobilization [86]. In line with this observation, miR-146b inhibition in free fatty acids-treated primary hepatocytes displays a decreased accumulation of cellular lipids [85]. Finally, miR-146 exerts a tumor suppressive role in liver cancer cells in vitro and in vivo, through the inhibition of AKT activation mediated by TRAF6 [87].

Hepatic expression of miR-34a is also regulated by CREB-regulated transcriptional activator 2 (CRTC2) [88]. Although no evidence of interaction or association between CRTC2 and PGC-1β has been so far reported, CRTC2 can positively regulate PGC-1α and downstream gluconeogenic genes in the liver [89]. We have discussed above that miR-34a is able to inhibit PGC-1α translation. Therefore, it seems that an intricate network between PGC-1s and miR-34a exists, probably in order to fine tune the metabolic pathways of the cells. Additionally, also p53 directly targets miR-34a, which in turn can activate p53 by downregulating SIRT1 action, thereby delineating a positive feedback loop. An increased miR-34a/SIRT1/p53 activation has been observed in the hepatocytes of CCl_4_-induced rat liver fibrosis, where it drives hepatocyte apoptosis, thus allowing the activation of hepatic stellate cells and perpetuation of fibrosis [90]. Notably, by inhibiting SIRT1, miR-34a can affect the functioning of a multitude of pathways. Indeed, increased miR-34a levels have been detected in liver of NAFLD animals and morbid patients [91,92,93,94], while miR-34a inhibition resulted in active SIRT1 and increased PPARα level, with concomitant restoration of triacylglycerol content and mitochondrial transmembrane potential in mice liver [94,95]. Moreover, the downregulation of miR-34a resulted in a lower expression of fibrotic genes upon alcohol-feeding in mice [96]. Indeed, miR-34a level is increased in the liver of ethanol fed rats and mice as well as in heavy drinkers [96,97]. Furthermore, the downregulation of SIRT1 induced by miR-34a leads to the activation of 3-hydroxy-3-methylglutaryl-CoA reductase (HMGCR), which boosts new cholesterol synthesis, thus promoting high liver injury and cardiovascular risk [92]. Interestingly, the ursodeoxicholic acid (UDCA), a secondary bile acid that contributes to cholesterol homeostasis by reducing the rate of intestinal cholesterol absorption, blocks miR-34a activity in the liver of NAFLD rats [91,98]. Moreover, in obesity conditions miR-34a attenuates fibroblast growth factor 19 (FGF19) signalling, a key pathway in the regulation of bile acids synthesis [99]. Although a direct involvement of PGC-1β in cholesterol and bile acids pathways has not been reported so far, it would be fascinating to explore this network in the light of the previously mentioned evidence. Notably, it has been reported that PGC-1α activates cholesterol 7-alpha-hydroxylase (CYP7A1), the key enzyme committed to the conversion of cholesterol into bile acids [100].

Despite the profibrotic role of miR-34a in NAFLD and alcoholic liver disease, several studies reported a tumor suppressive role of miR-34a in HCC, even if none of them deeply investigated miR-34a role in an in vivo cancer model [93,101,102].

Overall, PGC-1β overexpression leds to the upregulation of microRNAs that are extensively involved in liver pathophysiology. If most of them exert an anti-fibrotic role by blocking inflammatory pathways, others promote NAFLD and its sequelae. Notably, the PGC-1β overexpression in the liver has been found to protect against NAFLD and NASH, whereas it supports tumor growth and progression [103,104]. Therefore, it is possible that a fine tuning of these microRNAs upregulated in LivPGC-1β mice exists and that further studies are mandatory to understand this intricate network.

miR345-3p is upregulated in both LivPGC-1α and LivPGC-1β mice compared to wild-type. To date, no study focused on miR345-3p in the liver has been published. However, it has been observed that this microRNA is decreased in non-small cell lung cancer and it is enriched in mitochondria from failing heart at early stage [105,106]. Bioinformatic analysis revealed that the expression of miR-345-3p is associated with energy metabolism and oxidative stress [105]. Given that both coactivators have a similar effect on the induction of miR-345-3p, it would be interesting to understand if this microRNA plays a role in the common pathways regulated by PGC-1s in the liver, namely, mitochondrial biogenesis and/or antioxidant response.

From our analysis, only one microRNA was increased in LivPGC-1α mice, i.e., miR-150. Of note, this microRNA is located downstream of two genes whose expression products are known to interact with PGC-1s, namely Nitric Oxide Synthase Interacting Protein (Nosip) and Prmt1 (Table 1). However, to date no information regarding a direct regulation of these genes by PGC-1s has been described. Whole-body miR-150 ablation in mice prevents Fas-induced apoptosis by regulating AKT, a direct target of this miRNA [107]. Similarly, in HFD-induced NAFLD mice model and in NAFLD patients the levels of miR-150 are increased and miR-150 deficiency ameliorates hepatic steatosis and insulin resistance with concomitant reduction of fatty acids uptake and synthesis as well as of gluconeogenesis [108]. On the contrary, several works indicated that miR-150 inhibits Col4a4, Sp1 mediator of Col1a1, and α-smooth muscle actin (α-SMA), therefore playing an anti-fibrotic role in the liver [109,110]. Intriguingly, endoplasmic reticulum stress induces the expression of the unfolded protein response sensor IRE1α, which in turn mediates miR-150 degradation, thus enhancing liver fibrosis [109]. In line with these data, miR-150 is repressed in HCC, and its re-expression in hepatoma cells suppresses cancer cell proliferation, migration and invasion by blocking the matrix metalloproteinase 14 (MMP14) and the GAB1/EK axis [111]. Overall, these data would fit with PGC-1α role in protecting the liver from NAFLD, even if several controversies on the role of this coactivator on HCC still exist [18]. Probably, new analyses taking into account miR-150 will provide further insights into this scenario.

## 3. Materials and Methods

### 3.1. Animals

*LivPGC-1β* transgenic mice were previously generated in our laboratory by Dr. E. Bellafante [103]. *LivPGC-1α* transgenic mice were generated by injecting into the pronuclei of the fertilized eggs of the FVB/N mice the transgene digested with HpaI. To generate the pLiv.7 PGC-1α, first hPGC1α (2.4 kb) fragment was generated by PCR from pcDNA4-His-PGC1α plasmid (Addgene, Watertown, MA, USA). Then the fragment was subcloned KpnI and XhoI site of pLiv7 which carries the promoter, exon 1 and a fragment of exon 2 of the Apolipoprotein E, a protein expressed exclusively in the liver. Mice carrying the transgene were identified by PCR of genomic DNA to confirm the presence of an hPGC1α coding sequence. *LivPGC-1β* and *LivPGC-1α* mice display hepatic-specific PGC-1β or PGC-1α overexpression, respectively. We used wild-type (WT) FVB/N mice (referred as WT mice) as a control group. All mice were housed with a standard diet provided ad libitum and examined daily. Genotyping was performed using DNA extracted from tail biopsies of 4 weeks-old pups, and new breeding harems of 8 weeks-old mice were established to expand the population. The ethics committee of the University of Bari approved (1142/2016-PR and 509/2019-PR) this experimental setup, which was also certified by the Italian Ministry of Health in accordance with internationally accepted guidelines for animal care.

### 3.2. RNA Samples Preparation

Tissues were snap frozen in liquid nitrogen and stored at −80 °C. Total RNA was isolated by QIAzol reagent (Qiagen, Hilden, Germany) following the manufacture’s instruction, after tissue homogenization through Qiagen TissueLyser (Qiagen, Hilden, Germany). To avoid possible DNA contamination, RNA was treated with DNase (Thermo Fisher Scientific, Waltham, MA, USA). RNA purity and concentration were checked by spectrophotometer, while the RNA integrity was assessed by Bio-RadExperion™(Bio-Rad, Hercules, CA, USA). Only samples with Relative Quality Index (RQI) > N8 were used for microarray analysis. Samples were stored in aliquot at −80 °C prior to use.

### 3.3. Microarray Analysis of miRNA Expression Profiles

Microarray gene expression analysis was performed on RNA extracted from the mice liver. Whole RNA (400 ng) was studied using the Illumina Universal MicroRNA Expression Profiling (MicroRNA Expression profiling kit and Illumina Universal Bead Chips (Illumina, San Diego, CA, USA)). The bead chips were read on the Illumina iScan System microarray platform (San Diego, CA, USA). Upon the manufacturer instructions, data were processed using the Illumina Genome Studio Software (Illumina, San Diego, CA, USA) through specific algorithms of filtration and cleaning of the signal. All the items with detection *p* value > 0.01 were excluded. Data were normalized together with the quantile method. Background was not subtracted. Final output consisted of normalized fluorescence intensity of each probe (Average signal), representing the expression levels of each miRNA. This initial screening allowed us to identify 655 miRNAs to be exported from Genome Studio Software on which, after transformation log2-transformed data, we thus performed the statistical analysis. The identification of up- or down-regulated genes was performed by comparing miRNA expression of transgenic mouse group versus wild type mice at basal condition. Prediction of miRNA targets was performed using TargetScan [112]. The gene set enrichment analysis was performed using EnrichR [113,114].

### 3.4. Reverse Transcription and Real-Time Quantitative Polymerase Chain Reaction (qPCR) for miRNA Expression

For miRNA expression analysis, reverse transcription of 10 ng of RNA was performed using TaqMan microRNA Reverse Transcription Kit (Thermo Fisher Scientific, MA, USA), following the manufacturer’s instructions. Real-time qPCR assays were performed in 96-well optical reaction plates using a Quantum5 machine (Thermo Fisher Scientific, MA, USA); qPCR assays were conducted in triplicate wells for each sample, using pre-validated TaqMan Assays and TaqMan Universal Master Mix (Thermo Fisher Scientific, MA, USA), according to the manufacturer instructions. Quantitative normalization was performed using sno202 and sno234 as internal controls. Relative quantification was done using the ΔΔCT method.

### 3.5. Statistical Analysis

The results are expressed as mean ± standard error of the mean (SEM). All the statistical analyses were performed using GraphPad Prism software (v5.0; GraphPad Software Inc., San Diego, CA, USA). Comparisons of three groups were performed using a one-way analysis of variance (Anova) test followed by Tukey’s test for multiple comparison. At least *p*-value < 0.05 was considered statistically significant.

## 4. Conclusions

PGC-1α and PGC-1β exert divergent functions on liver metabolism and regulate different pathways in order to fulfil the cellular energetic demands and maintain whole-body homeostasis. The post-transcriptional modulation of PGC-1s, either by post-translational modifications and by microRNAs, allows a prompted response to both fed and fasted conditions. The failure of this regulation results in disease onset and progression. On the other hand, PGC-1s’ overexpression is associated with higher levels of several microRNAs involved in hepatic metabolic pathways, whose alteration may lead to the initiation of liver steatosis, NAFLD and HCC. Here we reported initial insights into PGC-1-driven regulation of miRNAs involved in liver metabolism, thus adding novel pieces in the puzzle of modulation of liver energetic and potentially offering targets for future treatment development.

## Figures and Tables

**Figure 1 ijms-20-05735-f001:**
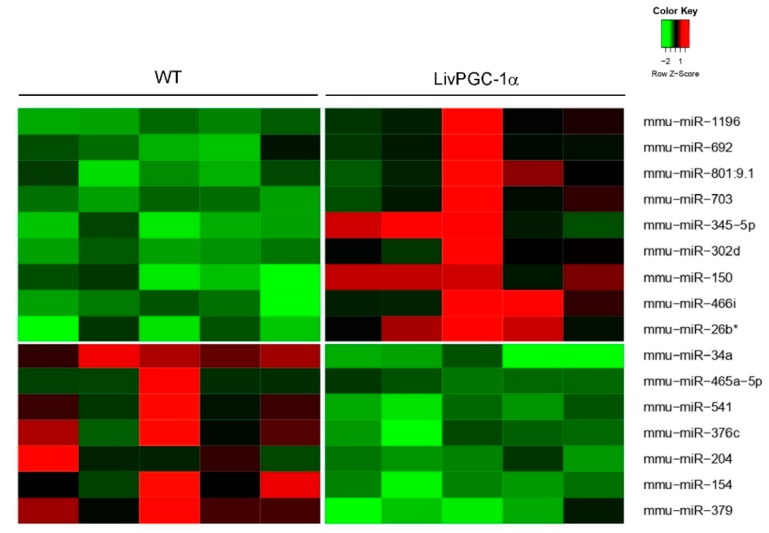
Microarray analysis of hepatic microRNAs in mice overexpressing proliferator-activated receptor (PPAR)-γ coactivator alpha (PGC-1α) in the liver compare to wild type control. The data are shown in a heatmap with a matrix format; each single row represents the expression of one microRNA in a single liver sample harvested from overnight fasted LivPGC-1α mice or WT controls (column). To visualize the results, the expression levels of each gene are represented by a color (red: expression greater than the mean; black: expression equal to the mean; green: expression smaller than the mean). The * is used to distinguish mature microRNA that, differently from the dominant-highly expressed microRNA, originates from the opposite arm of the hairpin precursor.

**Figure 2 ijms-20-05735-f002:**
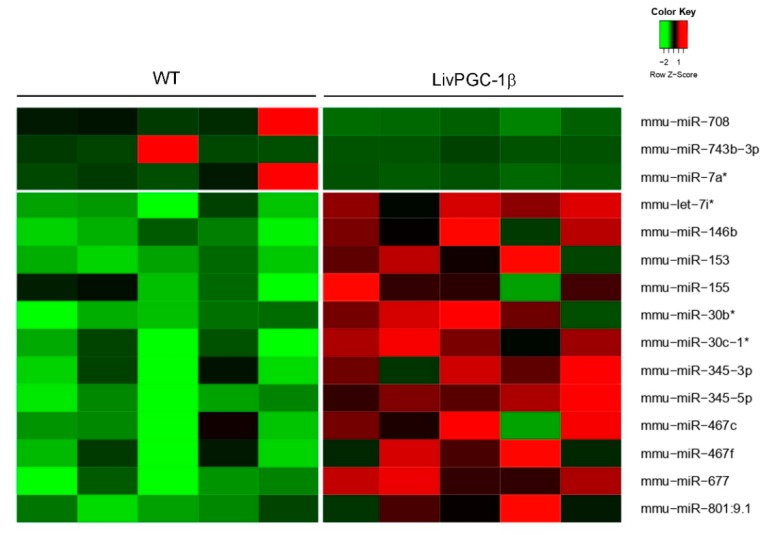
Microarray analysis of hepatic microRNAs in mice overexpressing PGC-1β in the liver compare to wild type control. The data are shown in a heatmap with a matrix format; each single row represents the expression of one microRNA in a single liver sample harvested from overnight fasted LivPGC-1β mice or WT controls (column). To visualize the results, the expression levels of each gene are represented by a color (red: expression greater than the mean; black: expression equal to the mean; green: expression smaller than the mean). The * is used to distinguish mature microRNA that, differently from the dominant-highly expressed microRNA, originates from the opposite arm of the hairpin precursor.

**Figure 3 ijms-20-05735-f003:**
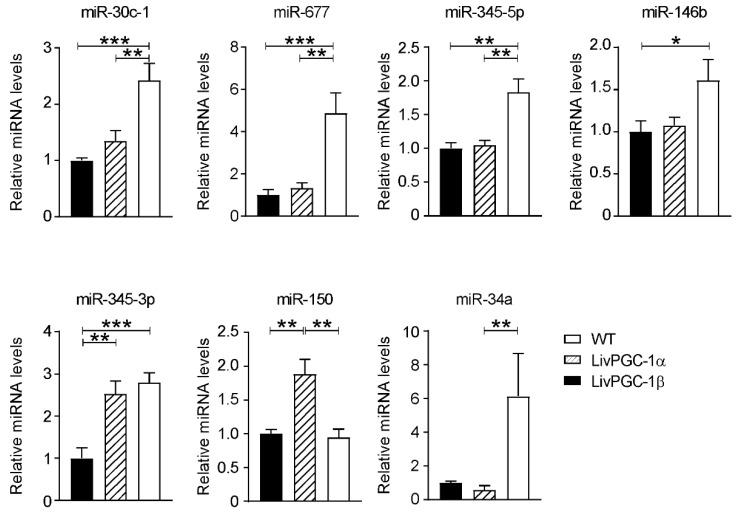
Real-time quantitative polymerase chain reaction (RT-qPCR) validation of hepatic microRNAs differentially modulated by overexpression of PGC-1α or PGC-1β specifically in the liver. Relative mRNA expression of microRNAs from liver samples collected from LivPGC-1α and LivPGC-1β mice compared to wild-type (WT) controls. All animals were sacrificed after overnight fasting. The data were normalized on the geometrical mean (“best-keeper” gene) of sno202 and sno234 levels, presented as relative expression values, and plotted as means ± standard error of the mean (SEM). Comparison of different groups (*n* = 6 mice/group) was performed using One-way ANOVA followed by Tukey’s test for multiple comparison. A *p*-value of 0.05 was considered significant (* *p* < 0.05; ** *p* < 0.01; *** *p* < 0.001).

**Figure 4 ijms-20-05735-f004:**
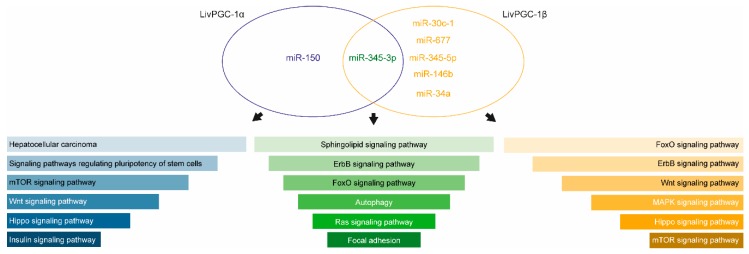
MicroRNA target prediction and enrichment analysis of the predicted target genes. The selective overexpression of PGC-1α or PGC-1β in the mice liver lead to the upregulation of different microRNAs. Genes targeted by differentially expressed miRNAs were predicted using TargetScan database. The gene set enrichment analysis was performed using EnrichR. The bar graphs display the predicted mouse Kyoto Encyclopedia of Genes and Genomes (KEGG) pathways based on *p*-value ranking.

**Table 1 ijms-20-05735-t001:** Genomic characterization of microRNA upregulated by Pgc-1s. Abbreviations: Nfyc, nuclear transcription factor Y subunit gamma; Atp5b, ATP synthase F1 subunit beta; Yy1, Yin Yang 1; Elovl3, ELOVL fatty acid elongase 3; Nfkb2, nuclear factor kappa B subunit 2; Nosip, nitric oxide synthase interacting protein; Prmt1, protein arginine methyltransferase 1; PPARα, peroxisome proliferator-activated receptor alpha.

miRNA	Sequence	Location	Proximal Gene	Association with Pgc-1
mmu-miR-30c-1	*cugggagaggguuguuuacucc*	chr4: 120769534-120769622	Inside *Nfyc* gene	Nfyc is a Pgc-1α target in striatal
mmu-miR-677	*uucagugaugauuagcuucuga*	chr10: 128085286-128085363	Inside *Atp5b*	Atp5p is a Pgc-1s target
mmu-miR-345-5p	*gcugaccccuaguccagugcuu*	chr12: 108836973-108837068	Downstream to *Yy1*	Yy1 is regulated by and regulates Pgc-1s
mmu-miR-345-3p	*ccugaacuaggggucuggagac*	chr12: 108836973-108837068	Downstream to *Yy1*	Yy1 is regulated by and regulates Pgc-1s
mmu-miR-146b	*ugagaacugaauuccauaggcu*	chr19: 46342762-46342870	Downstream to *Elovl3* and *Nfkb2*	Elovl3 is a PPARα target
mmu-miR-34a	*uggcagugucuuagcugguugu*	chr4: 150068454-150068555	Downstream to *Gm13067*	No information available
mmu-miR-150	*ucucccaacccuuguaccagug*	chr7: 45121757-45121821	Downstream to *Nosip* and *Prmt1*	Pgc-1s promote eNos expression, which in turn interact with Nosip. Prmt1 regulates Pgc-1s.

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
