# Peer review of "Hepatic MicroRNA Expression by PGC-1α and PGC-1β in the Mouse"

_ijms, 2019, doi:10.3390/ijms20225735_

Round 1
Reviewer 1 Report
The paper has been resubmitted with few changes. I understand it is a contribution to a thematic series, but I still consider its structure somewhat different from my view about standard types of papers. In fact, the paper is marked as Article on top.
There a number of typos and spelling mistakes. The use of commas is inadequate in some places, for instance, the use of comma before that. The use of capitals in many places, for instance Lysine and Arginine. In other places I understand capitals are intended to explain acronyms. ATPsynthase should be separated, while acetyltransferase should be a single word. Interacts in l. 116 should be the verbal form in plural.
I suggest a revision of the text to eliminate these flaws.
Author Response
Thank you for the opportunity to revise our paper once more for the International Journal of Molecular Science. We are grateful for the careful and considered comments and suggestions and have tried to respond to each. We believe these revisions have resulted in a significantly improved manuscript. Below, we outline how we have handled each of Reviewer’s comments. We reiterate each suggestion in italics.
REVIEWER 1
Comments and Suggestions for Authors
The paper has been resubmitted with few changes. I understand it is a contribution to a thematic series, but I still consider its structure somewhat different from my view about standard types of papers. In fact, the paper is marked as Article on top.
There a number of typos and spelling mistakes. The use of commas is inadequate in some places, for instance, the use of comma before that. The use of capitals in many places, for instance Lysine and Arginine. In other places I understand capitals are intended to explain acronyms. ATPsynthase should be separated, while acetyltransferase should be a single word. Interacts in l. 116 should be the verbal form in plural.
I suggest a revision of the text to eliminate these flaws.
We thank Reviewer 1 for the constructive comments on our manuscript. We have made every attempt to fully address these comments in the revised manuscript, and we believe these revisions have resulted in a significantly improved manuscript. We apologize for our inattention in the previous version of the paper. As suggested, we carefully looked throughout the manuscript and we corrected typos and spelling mistakes.
Reviewer 2 Report
This study still only ends with preliminary microarray analysis and real-time PCR verification. The detailed mechanism of how PGC1 regulates these miRNAs must be clarified. I feel that at this stage, this paper cannot meet the high standards of “International Journal of Molecular Sciences”.
Author Response
Thank you for the opportunity to revise our paper once more for the International Journal of Molecular Science. We are grateful for the careful and considered comments and suggestions and have tried to respond to each. We believe these revisions have resulted in a significantly improved manuscript. Below, we outline how we have handled each of Reviewer’s comments. We reiterate each suggestion in italics.
REVIEWER 2
This study still only ends with preliminary microarray analysis and real-time PCR verification. The detailed mechanism of how PGC1 regulates these miRNAs must be clarified. I feel that at this stage, this paper cannot meet the high standards of “International Journal of Molecular Sciences”.
We would like to thank Reviewer 2 for the effort and time he/she put in revising our manuscript. As previously clarified, our intention was to write a comprehensive review about microRNAs and PGC-1 in the liver. Unfortunately, up-to-date few data regarding microRNAs regulation by PGC-1 have been published. Therefore, here we presented new raw data about this topic in order to not only fulfill all the aspects of our topic, but also to guarantee the novelty of this review as respect to others in this field.
This manuscript is a resubmission of an earlier submission. The following is a list of the peer review reports and author responses from that submission.
Round 1
Reviewer 1 Report
This is an article with a non canonical structure. At first, I thought I was reading a review article structured in several sections, but this notion had to be modified when I found original experiments without a definite section devoted to Results and a Materials and Methods section at the end. The conclusions are not clear and the mention to the need of new studies appears at least four times lines 180, 388, 411, and 421.
The Introduction is very long and rather than devoted to set the scope where the paper is expected to contribute, it is highly speculative.
Finally, the conclusions are limited to describe the changes induced on the microRNA signature by overexpression of PGC-1α and PGC-1β. These animals show constitutive activity and I wonder whether they reflect the physiological function of the proteins versus a hyperfunctional state not fully fitting with current pathophysiological conditions
Reviewer 2 Report
PGC-1 is involved in the regulation of mitochondrial function, gluconeogenesis and lipid metabolism, and plays an important role in liver metabolism. PGC-1 is a powerful transcriptional coactivator that can interact with many transcriptional factors to regulate a variety of protein expressions. However, there are few studies investigating its regulation on microRNAs. In this study, the authors used animal models combined with microarray analysis to investigate microRNAs that may be regulated by PGC-1. The results showed that multiple microRNAs involved in liver metabolism were induced by PGC1, which may also indicate that PGC1 is involved in the regulation of liver metabolism. The results of this study are interesting and helpful in understanding the mechanism of liver related diseases. However, this study only ends with preliminary microarray analysis and real-time PCR verification. The detailed mechanism of how PGC1 regulates these miRNAs must be clarified. I feel that at this stage, this paper cannot meet the high standards of “International Journal of Molecular Sciences”.